# A Review of Gas Sensors for CO_2_ Based on Copper Oxides and Their Derivatives

**DOI:** 10.3390/s24175469

**Published:** 2024-08-23

**Authors:** Christian Maier, Larissa Egger, Anton Köck, Klaus Reichmann

**Affiliations:** 1Materials Center Leoben Forschung GmbH, Roseggerstrasse 12, 8700 Leoben, Austria; larissa.egger@mcl.at (L.E.); anton.koeck@mcl.at (A.K.); 2Institute for Chemistry and Technology of Materials, TU Graz, Stremayrgasse 9, 8010 Graz, Austria; k.reichmann@tugraz.at

**Keywords:** gas sensors, metal oxide, cupric oxide, cuprous oxide, nanomaterials, nanoparticles, carbon dioxide

## Abstract

Buildings worldwide are becoming more thermally insulated, and air circulation is being reduced to a minimum. As a result, measuring indoor air quality is important to prevent harmful concentrations of various gases that can lead to safety risks and health problems. To measure such gases, it is necessary to produce low-cost and low-power-consuming sensors. Researchers have been focusing on semiconducting metal oxide (SMOx) gas sensors that can be combined with intelligent technologies such as smart homes, smart phones or smart watches to enable gas sensing anywhere and at any time. As a type of SMOx, p-type gas sensors are promising candidates and have attracted more interest in recent years due to their excellent electrical properties and stability. This review paper gives a short overview of the main development of sensors based on copper oxides and their composites, highlighting their potential for detecting CO_2_ and the factors influencing their performance.

## 1. Introduction

Since the last century, efforts in the development of gas sensor devices have greatly increased because of new aspects in health science and stricter legal regulations. In 2022, the United Nations declared clean air as a human right that is only possible by reducing pollution. According to a 2023 WHO report on household air pollution [1], every year, 6.7 million people die from such pollution, which causes cardiovascular disease, respiratory illness and cancer. Therefore, indoor air quality (AQ) monitoring is very important to detect harmful or even toxic gases and substances like carbon monoxide (CO), carbon dioxide (CO_2_), volatile organic compounds (VOCs) and nitrogen oxides (NO_x_). In order to continuously monitor pollution indoors as well outdoors, very sensitive and low-power-consuming gas sensors are needed [2,3].

One of the most challenging toxic gases is carbon dioxide, which is a key indicator of climate change and clean air. Carbon dioxide is well known as a greenhouse gas, and its amount in the atmosphere has increased over the years. The main source is the burning of fossil fuels by industries and combustion engines [4]. In agriculture, CO_2_ is a byproduct of biogas synthesis and also indicates freezing damage in oranges. The main problem is caused by the storage of food in silos, where very high concentrations of CO_2_ accumulate. The food industry uses a modified atmosphere in packaging to keep food fresh for longer. Here, the loss of CO_2_ is a sign of leakage [5,6].

Since most of the population spends 90% of their time indoors, indoor air quality monitoring is also becoming a more important topic. Detecting CO_2_ is especially important, since increased indoor CO_2_ concentrations are caused by human respiration and reduced air exchange due to better insulation and tighter windows. High indoor concentrations of CO_2_ (Table 1) can lead to issues like increased breathing rate, disorientation and loss of consciousness [7]. In order to achieve controlled ventilation of rooms, which is necessary to alleviate high CO_2_ concentrations [8], low-energy-consuming and fast-responding sensors are essential. In order to achieve this goal, it is necessary to employ technologies such as artificial intelligence in combination with electronic noses. According to [9], the algorithms used for this purpose consist of three important parts. The first part is drift calibration, which is often performed using on-field calibration techniques such as the least squares method and the extremely randomized trees algorithm [10,11]. The second part includes signal pre-processing and data selection. The last part is the pattern recognition system, which is capable of recognizing odors.

The integration of highly efficient sensors in smart systems, such as wearables [14] and air quality monitors for smart homes [15], is enabled by nanotechnology, which offers several advantages in terms of performance. Nanostructures, such as nanoparticles and nanowires, have a very high surface-to-volume ratio, which results in increased sensitivity and selectivity [16]. Several companies have already commenced the production of commercial gas sensors based on nanotechnology [17]. In the case of CO_2_ gas sensing, organic, inorganic and hybrid materials are under investigation. Organic materials have many desirable properties such as chemical reactivity, mass transport and gas diffusion. On the other hand, inorganic materials are much more mechanically stable and have better optical properties and higher conductivity. The idea of combining organic and inorganic compounds led to research on hybrid materials [18]. Nevertheless, the development of inorganic materials is increasing because they are much more cost-effective. Metal oxides are promising candidates in this group. The ability to produce nanomaterials of different size, shape, morphology, thickness and porosity makes them promising materials for sensing. Their easy integration into micro-electromechanical systems (MEMS) technology using physical vapor deposition or screen printing could open the market for devices based on metal oxides [19].

## 2. Market Overview and Present Research

This section describes common CO_2_ gas sensors on the market. They can be classified by their measurement principle: optical, acoustic or electrochemical.

### 2.1. Optical Sensors

In the late 1930s, the first non-dispersive infrared (NDIR) sensors based on infrared spectroscopy were developed in the United States. Subsequently, in 1955, this technology could measure more than 30 different gases [20]. This was the starting point for technology that could detect potentially harmful gases based on infrared light. These optical sensors are most commonly used for the detection of gases such as carbon monoxide (CO), carbon dioxide (CO_2_), sulfur dioxide (SO_2_), nitrogen oxides (NO_x_), nitrous oxide (N_2_O), ammonia (NH_3_), hydrogen chloride (HCl), hydrogen fluoride (HF) and methane (CH_4_) [21]. The main advantage of optical sensors is their ease of operation in the absence of oxygen, which allows them to be used in special or inert atmospheres. They can be used over a wide monitoring range. On the other hand, the disadvantages are problems with light interference and the fact that not all gases absorb light [22]. The majority (83%) of CO_2_ sensors on the global market are non-dispersive infrared sensors [23]. A typical NDIR sensor is composed of a light source (conventionally, a microbulb), a gas chamber with a reflective gold coating and a detector (pyroelectric or thermopile). The measurement principle is based on the absorption of light by CO_2_ and can be described by the Beer–Lambert law (Equation (1)) [24]:(1)$$\log_{10}\left(\frac{I_{0}}{I}\right)=\in *\;c*L$$

Here, I_0_/I is the ratio of the intensity of incoming and outgoing light, L is the length of the gas chamber or the interaction length, C is the gas concentration and ∈ is the absorption coefficient. The wavelength of the maximum absorption of CO_2_ is around 4.26 µm. Consequently, the NDIR sensor is operated in the mid-IR region (2.5 to 14 µm), where the adsorption is based on molecular vibration and rotations. The reduction in light intensity is proportional to the concentration of CO_2_. The main advantages of NDIR are the very high absorption coefficient of CO_2_ and the high gas specificity [23,25,26]. Compared to other sensor technologies, commercially available NDIR sensors can be operated between −40 and 70 °C [27]. There are drawbacks with all absorption-based methods, such as water vapor interference and detection limit (>30 ppm CO_2_) [21]. Miniaturization is limited because of the high absorption coefficient of CO_2_, since an optical pathway of a few centimeters is needed for detection [23].

Another type of optical sensor, the surface plasmon sensor, is based on the surface plasmon resonance effect, which can be described as the interaction of light with a material. The most widely used configuration is called the Kretschmann configuration, in which the light hits the base of a prism that is in contact with a metal. The electrons on the surface get excited by the light and start travelling on the surface. These electrons are called surface plasmons. One part of the light is absorbed by the material and the other part is totally reflected at an angle called a resonance angle. This angle depends on the refractive index of the investigated surface and is changed by the interaction of CO_2_ with the surface. This effect is only observed with materials that have a negative real part and small imaginary part of relative permittivity, such as Cu, Al, Au and Ag. This simple configuration is cost-efficient, and the sensor has high sensitivity. The fields of application are mainly in pharmacy and bio-sensing [18,28]. The drawbacks include the binding of certain gases on the surface and interference from the refractive index, which limit the sensor’s applicability [29,30].

The last type of optical sensor that is under investigation for CO_2_ detection is the optical fiber sensor. This type of sensor is made of a silica gel or polymer fiber covered by a material with a lower refractive index (e.g., polymer or sol–gel groups) [31]. The measurement principle is determined based on the method used, such as absorption, change of refractive index or fluorescence. Fluorescence-based measurement is used to detect CO_2_. Here, the intensity of the emission peak is measured, which is then used to calculate the concentration of the gas [32]. The advantages of fluorescence optical fiber sensors are the possibility of miniaturization, high sensitivity and applicability [31]. The disadvantages are toxicity issues, interference and a short lifetime [33].

### 2.2. Acoustic Sensor

Another type of CO_2_ sensor is based on changes in the resonance frequency of surface acoustic waves of the sensing material. Such a sensor is, therefore, called a surface acoustic wave (SAW) sensor. The device consists of a piezoelectric substrate with inter-digital transducers (IDTs). The substrate is often a single quartz or lithium niobate crystal, which is functionalized with a coating material, typically a polymer. The inter-digital transducer on one side of the substrate converts incoming electromagnetic signals into mechanical waves travelling on the substrate surface. The IDT on the other side of the substrate converts mechanical surface waves into electromagnetic signals. The adsorption of a gas on the surface causes a change in the mass or conductivity of the coating material. This changes the mechanical waves on the substrate, which can be seen at the electrical output signal [8,34]. The main challenge of this type of sensor is in fabricating a coating material that is energy-efficient and sensitive to only one type of gas [35]. Nevertheless, the easy technological compatibility and possibility of miniaturization make it an interesting research topic for humidity and gas sensors [36].

### 2.3. Electrochemical Sensors

The last type of sensor we will discuss is known as a solid-state electrolyte sensor and is the most promising candidate to use with MEMS and nanotechnology for CO_2_ gas sensing [5]. This type can further be divided into potentiometric, amperometric and resistive sensors based on the measurement method. Additional methods include measuring changes in the work function, capacitance or impedance. The gas measurement is based on changes in electrical signals caused by the reaction of gas molecules with the sensing layer, which is in most cases a semiconducting metal oxide. SMOx-based sensors usually require an operating temperature up to 700 °C [37]. Advanced micro-hotplate-based sensor devices (size: 74 × 74 µm^2^) [38] such as those described in [39] enable well-defined localized heating of only the sensing area; thus, they can be implemented in smart devices. The use of metal oxides has advantages, such as very high sensitivity and rapid response and recovery times. On the other hand, there are still challenges, such as a lack of selectivity, which is the ability to identify a single gas component in a gas mixture. Another issue is that the sensor response depends on the humidity and operating temperature (up to 300 °C) [40].

The first of this type of sensor is the potentiometric sensor, which measures the potential difference between a working electrode and a reference electrode. The presence of a gas changes the potential of the working electrode. The lambda sensor, also called a lambda probe or oxygen sensor, in the automotive industry is a prominent example. There are also many sensors being researched for the measurement of CO_2_ [41]. Sensors based on Na super ionic conductors (NASICONs), with the formula Na_1+x_Zr_2_Si_x_P_3−x_O_12_, are promising due to their high chemical stability and high conductivity at low temperatures [42].

Amperometric sensors measure changes in current. A chemical reaction on the surface of the working electrode causes a current flow [43]. The resulting current is measured and is directly proportional to the concentration of the target gas. This type of sensor is used in patient monitoring of O_2_ and CO_2_ and in fire detection [44]. CO_2_ detection with this type of sensor is possible by the use of NASICON or yttria-stabilized zirconia (YSZ) materials [41,45].

Another type is resistive sensors, which consist of a sensing layer contacted by two electrodes. DC voltage is applied between the two electrodes and the resistance is determined [44]. The change in resistance is due to contact with the target gas. Semiconductors are produced with many metal oxides and combinations thereof, such as SnO_2_ [46], ZnO [47], TiO_2_ [48] and BaTiO_3_ [49], which show changes in resistance by their interaction with CO_2_ [50].

Work function sensors are based on the Kelvin probe. These sensors measure changes in the contact potential difference between the sensing layer and the reference electrode. These changes are caused by changes in the work function of the sensing layer. The use of, e.g., CuO nanoparticles (NPs) as the sensing material makes the detection of CO_2_ possible [51].

Capacitive sensors measure the ability to store charges on the substrate. The interaction of gas molecules with the dielectric layer, which is in contact with two conductive layers, causes changes in the capacitance of the device. The use of polymers like polyimides or cellulose acetates makes capacitive sensors attractive for humidity sensing. Silanes such as 3-amino-propyl-trimethoxysilane and propyl-trimethoxysilane [52] or metal oxides can be used for the detection of CO_2_. The metal oxides that can be used for these sensors include CuO/BaTiO_3_, PbO, NiO and ZrO_2_ [52].

Another type of gas sensor is based on impedance analysis. A sinusoidal voltage with a small amplitude is applied to the sensor material and the resulting current is measured. The impedance is calculated as the ratio of voltage to current in the frequency domain and is described as a complex number. The information of the real part contains the ohmic resistance. The imaginary part consists of the capacitance and the inductance. Evaluating the impedance gives information about grain boundary resistance/capacitance, bulk resistance/capacitance and mass transport and chemical reaction rates. Such impedance sensors are under research for nitrogen oxides, hydrocarbons, carbon monoxide and water vapor [53,54]. In_2_O_3_ and BaTiO_3_ have been shown to have good properties for use in impedance measurements for CO_2_ sensing [55].

The CuO-based sensor is a type of electrochemical device that is currently the subject of considerable research and investigation. This is due to the fact that CuO nanostructures of varying dimensions can be easily fabricated on CMOS-integrated micro-hotplate chips [56]. The on-chip synthesis of nanostructures of other materials, such as SnO_2_ or ZnO, is constrained by the inherent difficulties associated with their fabrication at elevated temperatures [57]. One significant advantage of CuO nanostructures over other SMOx sensors is their independence from relative humidity, as evidenced by their consistent sensor response [58]. Additionally, the decoration of CuO nanostructures with nanoparticles, such as Pd for CO [59], has been successfully employed to enhance the sensor response to gases. Steinhauer’s review [60] provides a summary of the reactions observed between a variety of test gases and pure CuO.

## 3. Gas Sensing Mechanism

Many published papers are on n-type semiconductors (e.g., SnO_2_, ZnO, TiO_2_, WO_3_, In_2_O_3_ and Fe_2_O_3_) [61]. On the contrary, only a few published papers (less than 10%) deal with p-type semiconductors (e.g., WO_3_, Co_3_O_4_, Cr_2_O_3_, NiO, TeO_2_ and CuO) [61,62]. In general, the difference between these two types is that the charge carriers are electrons in n-type semiconductors and holes in p-type semiconductors. In both types, the sensing mechanism is based on the adsorption of a target gas, which causes a charge transfer that can be measured [63]. The mechanism in air depends on the oxygen adsorption of the metal oxide surface. Oxygen atoms have a very high electron affinity, and if they are adsorbed on the surface, they attract electrons from the valence band of the metal oxide [64]. The oxygen ions stay adsorbed on the surface and the species, molecular ($O_{2}^{-}$) or atomic ($O^{-}$,${\;O}^{2-}$), depends on the temperature. At temperatures below 100 °C, oxygen molecules take just one electron. If the temperature reaches 100 °C, oxygen molecules split into two oxygen ions by again taking one electron to form the $O^{-}$ species. At temperatures higher than 300 °C, oxygen ions again take one electron, which results in the $O^{2-}$ species [63,65,66].
(2)$$O_{2}+e^{-}=O_{2}^{-}\;(<100\;{}^{\circ}C)$$(3)$${\;O}_{2}^{-}+e^{-}=2O^{-}\;(100–300\;{}^{\circ}C)$$(4)$$O^{-}+e^{-}=O^{2-}\;(>300\;{}^{\circ}C)$$

The transfer of electrons from the p-type metal oxide caused by the adsorption of charged oxygen on the surface leads to the formation of a hole accumulation layer close to the surface, as illustrated in Figure 1. This accumulation layer, which contains holes as charge carriers, covers the whole surface and has low resistance. The core of the metal oxide is insulating, resulting in very high resistance [62,64]. The electrical behavior can be described by an electronic core–shell configuration This means that the electrical resistance across a polycrystalline p-type metal oxide is determined by a parallel connection between two resistors, (i) an insulating core and (ii) a low-resistance hole accumulation layer, and a serial connection across the grain boundaries [67]. If the p-type metal oxide is exposed to a gas, surface reactions cause an increase or reduction in the charge carriers in the hole accumulation layer. Therefore, the change in resistance of the hole accumulation layer is the determinable parameter in resistive-based gas sensing [62].

In addition, the band model (Figure 2) is often used to describe the electrical behavior of metal oxide semiconductors in gas sensing. Under inert conditions (Figure 2a), the bands are flat and the surface state is completely unfilled. As described before, chemisorption of oxygen happens when the metal oxide comes into contact with air [66]. The oxygen takes electrons from the valance band, which leads to trapping of electrons on the surface. For p-type semiconductors, the trapping of electrons near the surface increases the free charge carrier concentration of holes. In the band model, this effect can be described as an upward bending of the band that forms a hole accumulation layer close to the surface (Figure 2b). This formation causes a layer with low resistance; therefore, the resistance of the metal oxide decreases compared to that in the flat situation. A reducing gas will react with the oxygen on the surface, releasing electrons and causing a recombination with the free holes. Therefore, the number of free charge carriers is reduced, which narrows the hole accumulation layer [68]. The resistance of the p-semiconductor increases, which is shown in the band model (Figure 2c) as a decrease in band bending (q∆Vs). In contrast, an oxidizing gas will react with the oxygen by taking electrons due to its high electron affinity. This means that the trapped electrons on the surface are removed, which leads to growth of the hole accumulation layer and higher upward bending of the band. The resistance of the p-material is decreased. Both processes with reducing/oxidizing gas should be reversible [69].

Due to the fact that all metal oxide-based gas sensors react to oxygen, they are highly influenced by the humidity of the environment. It has been observed that large amounts of water will split into protons (H^+^) or hydroxyl ions (OH^−^) on the surface of these sensors. A closer look at the surface reaction can explain this behavior (Equation (5)). At low humidity levels, water molecules react with adsorbed oxygen ($O_{\mathrm{ad}}^{-}$) by the consumption of holes (h^+^) to form terminal hydroxyl groups $\left({\mathrm{Cu}}_{\mathrm{Cu}}^{+}-{\mathrm{OH}}^{-}\right)$ and free surface sites for oxygen adsorption.
(5)$$O_{\mathrm{ad}}^{-}+H_{2}O_{\mathrm{gas}}+2{\mathrm{Cu}}_{\mathrm{Cu}}+h^{+}↔2({\mathrm{Cu}}_{\mathrm{Cu}}^{+}-{\mathrm{OH}}^{-})+S$$

Therefore, the reaction of water vapor with the adsorbed oxygen reduces the band bending and increases the resistance. The electron affinity is also increased by the dipolar field, making it much harder for the electrons to leave the metal oxide [70].

At very high humidity levels, all active sites are occupied by protons and hydroxyl groups. This leads to the physisorption of other water molecules and the formation of a new layer. These water molecules can react with the chemisorbed protons, which results in hydronium ions (H_3_O^+^). However, the hydronium ions are also charge carriers that change the baseline resistance by proton hopping. Another effect of high humidity is competition between water molecules and gas molecules to reach active sites, which can also reduce the gas adsorption [69].

The remainder of this review discusses the resistive, impedance and work function types of gas sensors that make up the majority of CuO-based SMO_x_ sensors.

## 4. Characterization of Sensor Performance

This section presents the main performance parameters used to compare gas sensors, including sensor response and changes in work function. Other parameters, such as response time and recovery time, are also important but are not treated in this paper. There is no standardized rule in the literature for calculating sensor response or sensitivity. This paper uses the following equations to express the sensor response. In the literature, it is found that some papers also use these equations to calculate sensitivity, but according to the IUPAC definition [71], sensitivity is understood as the slope of the calibration curve dependent on the concentration or amount of analyte.

### 4.1. Sensor Response

Sensor response can be understood as a change in a measured property, such as resistance, current, capacity or impedance. Several factors affect sensor response, including composition, relative humidity, temperature of the sensing material, doping with additives, morphology and structure [72].

Two equations are used in the literature to express the sensor response as determined by resistance. The first one (Equation (6)) is calculated as the ratio between the resistance during gas exposure (R_g_) and the resistance in air (R_a_) [73]:(6)$$S=R_{g}/R_{a}$$

The second calculation (Equation (7)) determines the absolute change in resistance $\left|\Delta R\right|$ and calculates the ratio with the resistance in air (this value is often expressed as a percentage by multiplying it by 100) [74]:(7)$$S=\frac{\left|\Delta R\right|}{R_{a}}*100\%=\frac{R_{g}-R_{a}}{R_{a}}*100\%$$

Some authors, instead of using resistance, use the current (Equation (8)) [75], capacitance (Equation (9)) [76] or impedance (Equation (10)) [77] to determine the sensor response of the sensing layer:(8)$$S=\frac{I_{g}-I_{a}}{I_{a}}$$
(9)$$S=\frac{C_{g}-C_{a}}{C_{a}}$$
(10)$$S=\frac{Z_{g}-Z_{a}}{Z_{a}}$$

### 4.2. Change in Work Function

To determine the work function, in most cases, the Kelvin probe setup is used. In general, the contact potential is understood as the difference in work function between the two investigated materials [78]. Therefore, the change in work function (ΔΦ) is linked to the contact potential difference (CPD), where e is the charge of an electron (Equation (11)) [79]:(11)$$\mathrm{CPD}=\frac{1}{e}*\Delta \Phi$$

## 5. Cu Oxides

In general, copper has five oxides: cupric oxide (CuO), copper peroxide (CuO_2_), cuprous oxide (Cu_2_O), copper (III) oxide (Cu_2_O_3_) and paramelaconite (Cu_4_O_3_). Only CuO and Cu_2_O are industrially used, because the others are not stable or are difficult to synthesize [80,81,82]. These two copper oxides are both p-type semiconductors with good electrical properties that enable a large variety of possible applications, such as in lithium-ion batteries and supercapacitors [83]. Copper oxides are also widely used in catalysis and ceramics and as an electrode material due to their mechanical, catalytic and optical properties. The optical properties and the bandgap energy in the range of 1.2–2.1 eV make CuO interesting for use in solar cells [80,84]. The main advantages of using copper oxides in gas sensing are the low cost, non-toxicity and the possibility of fabricating different nanostructures [85,86].

### 5.1. Structural Configuration

Various structural configurations for CuO as a gas sensing material are reported in the literature, ranging from films with micrometer thickness down to small nanowires or nanoparticles. In addition, other nanostructures such as 1D nanorods, 2D nanosheets and 3D nanoflowers are considered to be good candidates for gas sensing application due to their high surface area [83]. There are also other morphologies (nanopatterns, hollow spheres, nanocrystals, nanobelts, nanospindles, nanocages, nanowalls and nanoribbons) and combinations of Cu_2_O/CuO that can be used for the detection of gases like VOCs, H_2_S, CO, CO_2_, H_2_ and NO_2_ [60]. Volanti et al. [87] reported that urchin-like CuO structures showed increased performance for sensing CO and NO_2_ compared to CuO fibers or nanorods. Structures such as nanocubes have a lower sensor performance in detecting CO than nanotubes. The morphology, therefore, has a strong influence on the gas sensing behavior [88].

### 5.2. Deposition Techniques for Copper Oxides

This section discusses the common deposition techniques for copper oxides, which are reported in [80] for CO_2_ gas sensing. Fabricating thin films includes adsorption on the surface, surface diffusion, incorporation at the surface structure and desorption of by-products.

#### 5.2.1. Spray Pyrolysis

Spray pyrolysis is a simple deposition technique that does not require vacuum conditions. The materials typical consist of a solution containing a precursor, an atomizer and a heated substrate. The precursor solution, which contains a salt of the material, is atomized to make small droplets. A list of typical precursors for the fabrication of CuO films is given in Table 2. After the heated substrate is reached, a chemical reaction takes place, resulting in the formation of a thin film at the surface [89].

Spray pyrolysis is controlled by many parameters, including aerosol generation, solvent evaporation, precursor decomposition, droplet impact and temperature. Temperature is the most important factor influencing the film morphology. The microstructure determines the electrical and optical properties. The properties of the film are also influenced by the precursor solution, the type of salt, the concentration and additives. Another factor that must be considered is the aerosol generation and transport. The generation of the aerosol starts at the atomizer, and two typical modes are used: in cone-jet mode, a Taylor cone of the liquid is formed at the tip of the nozzle, and in multi-jet mode, one jet is split into multiple smaller jets. For aerosol transport, it is necessary that no powder or salt products are formed. During transport to the heated substrate, the solvent evaporates, which leads to smaller droplets [93]. The type of film that forms depends on the distance to the substrate, where the intermediate reaction happens, as illustrated in Figure 3. If the distance to where the chemical reaction happens is large (Figure 3a), the solvent evaporates and a precipitate reaches the surface. After that, the precipitate decomposes and forms a porous film or, in the worst case, cracks can occur. The vapor reaches the surface and a dense film is formed by means of chemical vapor deposition (Figure 3b). At high temperatures (Figure 3c), the formation of particles happens in the vapor phase, and a powder film is formed on the surface. All of these parameters need to be considered when using spray pyrolysis as the deposition technique [94].

#### 5.2.2. Sputtering

The sputtering technique uses a bombardment of highly energetic ions on a target material. The collisions with the surface of the target material kick out atoms from the material, creating a vapor that diffuses toward the substrate to make a thin film. The system consists of a chamber that is used under vacuum conditions, and a working gas, such as Ar, is introduced. A voltage source is connected to the target material (−), and the ionized atoms (Ar^+^) hit the surface of the target. The knocked-out target atoms reach the substrate and produce a film on the surface. Depending on the voltage source, the sputtering can be direct current (DC) or radio frequency (RF) sputtering. The main difference is that in RF sputtering, the electrons are oscillating, and larger ions cannot follow the electric field. This causes the same effect as DC sputtering, i.e., the ions are accelerated to the negatively charged target. To improve the sputtering performance of both DC and RF, a magnetic field (Figure 4) can be applied near the target material to control the pathways of the electrons [96]. The movement of the ions is not influenced by the magnetic field. Therefore, only the electrons are trapped in the region close to the target. This is called the ionization region, where the electron density is much higher, resulting in a stronger plasma. This leads to increased ionization of the ions, which makes a higher sputtering rate possible [97]. Oxide films such as CuO films are fabricated by using a Cu target and adding oxygen into the vacuum chamber [98]. This special technique is called reactive sputtering.

#### 5.2.3. Thermal Evaporation

In thermal evaporation, the material is melted and evaporated by an electrical resistance heater, where a high direct current is applied (Figure 5). Another variant of this technique is heating by an electron beam. Resistance heating is an easier method but can be only used for materials with a low melting point. Often, a boat made of quartz, graphite, alumina, beryllia, boron nitride or zirconia containing the solid material can be used for the melting process. Otherwise, a solid bar can be heated by a resistive coil, which is called filament evaporation. The vapors of the target material condense on the colder substrate to form a solid thin film. A high vacuum below 1 × 10^−6^ torr (1.3 × 10^−4^ Pa) is necessary to avoid unwanted reactions or scattering by the gas phase [100,101].

After the deposition of Cu film, a thermal oxidation step is necessary. Here, temperature is the main influencing parameter, which determines the phase of the oxide. In a temperature range of 220–250 °C, Cu_2_O is formed. At around 350 °C, a mixture of Cu_2_O and CuO is formed. When the temperature is increased to 500 °C, a single CuO phase is obtained [103]. The reaction equation is as follows [104]:(12)$$2\mathrm{Cu}+\frac{1}{2}O_{2}\to {\mathrm{Cu}}_{2}O$$
(13)$${\mathrm{Cu}}_{2}O+\frac{1}{2}O_{2}\to \mathrm{CuO}$$

#### 5.2.4. Screen Printing

This method is primarily used to produce thick film metal oxide gas sensors. In the first step, a paste is fabricated containing the metal oxide and organic binders and solvents like carboxymethylcellulose and terpineol. This paste is then pressed onto the substrate through a screen mask with the desired shape. The sensing layer is dried at room temperature or by a controlled drying process. During the debinding process, the binder and organic residuals are removed. A high-temperature sintering step is required to form a stable and adherent gas sensing layer. The combination of BaTiO_3_ with CuO is often deposited by the screen printing method. The advantages of screen printing are its cost-effectiveness and the possibility of mass production with consistent performance [49,105,106,107].

#### 5.2.5. Hydrothermal Method

The hydrothermal method is chemical synthesis in an aqueous medium, which needs high temperature and more than 1 bar. The synthesis involves a crystal growing in heated water above the boiling point and high pressure in a steel vessel called an autoclave [108]. Many different materials, such as microporous materials and nanomaterials, can be fabricated [108,109]. In the case of oxide nanomaterials, the synthesis of nanoparticles, nanorods and sheets is reported in [110]. Other important factors that affect the reaction include the solvent, stabilizing agents, precursor concentration and reaction time. The main advantage of this one-step synthesis is that, due to the drastic reaction conditions, no post annealing is required. The method also leads to very high crystallinity [111]. However, the sensing behavior and properties of CuO nanostructures strongly depend on the morphology, size and crystallinity. As reported in [112], shape-controlling substances such as surfactants are often needed to achieve the desired structure. One disadvantage of using such substances is that the active sites of the CuO can be influenced by residual atoms of the surfactant. Studies have been conducted on surfactant-free methods for synthesizing spherical and flower-like CuO that exhibit favorable sensing properties [110]. There are many different starting materials for CuO, including copper sulfate, copper acetate and copper chloride [113].

#### 5.2.6. Sol–Gel

Another chemical method used to fabricate nanomaterials is the sol–gel technique, and an overview is given in Figure 6. This process provides good control over the surface and texture properties. The synthesis is a five-step process, starting with the mixing of precursors and solvents. In the first step, called hydrolysis, the metal alkoxide reacts with the solvent, either alcohol or water, and ends up in a metal hydroxide, as shown in Equation (14) [114]:(14)$$M-\mathrm{OR}+H_{2}O\to \mathrm{MOH}+\mathrm{ROH}\to (\mathrm{hydrolysis})$$

The sol is formed during the condensation step, when the solvent is eliminated to link the metal oxides together, as described by Equation (15). This combination of metal oxides leads to the growth of a polymeric network [114].
(15)MOH+XO−M→MOM+XOH→(condensation)M = metal, X = H or alkyl Group (C_n_H_2n+1_)

The growth of this network increases the viscosity during the formation of a connected porous structure; therefore, it is called a gel. The third step, the aging process, continues with condensation in the liquid phase, which results in structural changes and decreases the porosity. The next step is the drying process, which can be performed in three ways: thermal, supercritical or freeze drying. Thermal drying causes a reduction in total volume, which results in decreased surface area and pores. This structural collapse results in a product named xerogel. In contrast, supercritical drying does not change the structure and leads to a large number of pores and increased surface area. The third process is carried out by freezing the gel to form a cryogel, which contains fewer pores compared to thermal drying. The last step to obtain the final product is calcination, by which organic residuals are removed. The size and density of the pores are controlled by the temperature [116].

For the fabrication of copper oxide films, copper (II) acetate in ethanol [117] or cupric chloride in methanol can be used as the starting materials. It is also possible to control the formation of CuO or Cu_2_O by temperature, as shown in Equations (16) and (17) [118]:(16)$$4{\mathrm{CuCl}}_{2}+4H_{2}O\overset{360\;{}^{\circ}C\;}{\to }2{\mathrm{Cu}}_{2}O+8\mathrm{HCl}+O_{2}$$
(17)$${\mathrm{CuCl}}_{2}+H_{2}O\overset{400-500\;{}^{\circ}C}{\to }\mathrm{CuO}+\mathrm{HCl}$$

### 5.3. Doping

In order to enhance the sensor performance, additives are used to reduce the energy required for the chemisorption of gas molecules on the surface. In the majority of published works, these additives are referred to as dopants [119]. There is a notable distinction between the use of noble metals and metal oxides as dopants. Two primary forms of doping exist: surface loading and bulk integration [120].

#### 5.3.1. Surface Loading

In the first approach, illustrated in Figure 7, a small amount of an additive is dispersed over the surface of the metal oxide. The additive has a high surface-to-volume ratio and forms a separate phase at the surface. The particle size of the additive is much smaller than the grain size of the metal oxide. The term “loaded surface” refers to the metal oxide being coated with the additive. This combination of a metal oxide with an additive can be explained as the formation of nano-Schottky barriers between the two. The additive provides a negative charge on the surface that works under gas exposure as a region with high conductance. At the same time, small clusters form to provide active sites for the adsorption or desorption of the target gas molecules. Examples include Au, Pd and Pt, which enhance sensing performance in the form of lower operating temperature or higher sensitivity or selectivity [120,121,122].

Influence of noble metals as dopants

There are many papers on SMO_x_ sensors that discuss the influence of doping [123,124,125]. The most commonly used dopants are noble metals such as palladium, platinum, silver and gold [123]. These dopants are used as nanoparticles and deposited on the surface by means of various techniques. The nanoparticles change the sensitivity and selectivity and reduce the operating temperature of the sensor. These effects are described by adsorption and catalytic effects. Two mechanisms can take place, as illustrated in Figure 8.

The first mechanism is chemical sensitization (Figure 8a), in which the noble metal (e.g., Pt) acts as a promotor and activates gas molecules for further reactions. The activated gas molecules migrate (spill over) and react with the adsorbed oxygen on the surface. This reduces the negatively charged oxygen on the surface, which then causes a change in the conductivity of the metal oxide [119,126].

The second mechanism is based on an electron exchange between the metal oxide and the noble metal (Figure 8b). Here, the noble metal acts as an electron donor or acceptor. The connection of a metal oxide with a noble metal causes a bending of the conduction band similar to the gas molecules (as shown in Figure 2b), which results in the formation of a Schottky barrier. The electrons are moved from the conduction band of the metal oxide to the noble metal, which leads to the formation of an interface dipole layer. This barrier (electron-depleted space charge layer) near the surface does not allow the recombination of electrons and holes, which increases the gas response [123]. With this type of mechanism, it is essential to use a noble metal that can form oxides in air. Pt does not form oxides, but Ag or Pd can easily do so. The reaction of a reducing gas with a noble metal changes the oxidation state of the noble metal and causes a charge transfer to or from the metal oxide [61,123].

Influence of metal oxides as dopants

The combination of two metal oxides forms an interface, called a heterojunction, which enhances the performance. The difference in Fermi levels between the two metal oxides causes an electron flow from the higher to the lower Fermi level until an equilibrium is reached, and this forms the electron depletion or hole accumulation layer. [124]. Heterojunctions can be subdivided into p–n, n–p, n–n and p–p junctions. With p–n and n–p junctions, the electrons flow from the n-type to the p-type, while the holes travel in the opposite direction. A depletion layer is formed at the interface that reacts under gas exposure, with a larger response than the n-type material [127]. With an n–n heterojunction, the electrons from the n-type flow from the higher Fermi level to the lower Fermi level. A depletion layer forms at the n-type with the higher Fermi level and an accumulation layer at the n-type with the lower Fermi level. In contrast, a p–p heterojunction causes hole movement from the p-type with the higher valence band level to the lower level. The p-type with the higher valence band level forms a depletion layer and the p-type with the lower valence band level forms an accumulation layer [123]. For all types of heterojunction, a higher sensor performance is achieved and the sensing layer incorporates, mainly, the advantages of both metal oxides [127].

#### 5.3.2. Bulk Integration

The second method is integrating additives into the bulk (Figure 9). In general, there are two options for bulk integration (Figure 9a): incorporating the metal oxide at the surface or in the bulk, which, in either case, will form a metal oxide matrix. In contrast to surface loading, the additives do not make a separate phase on the surface. If the grain size of the additive and the metal oxide are similar, then it refers to bulk integration. If the grain size of the additive is smaller than that of the metal oxide, it refers to surface loading [121]. In the case of bulk integration, the metal oxide and additive can be arranged randomly (Figure 9b) or in order (Figure 9c). This doping of the bulk changes the crystal structure of the metal oxide, can create new acceptor and donor states and can shift the bulk or surface Fermi level [120]. The conduction pathway of the bulk depends either on the dominant species with low resistance or a heterojunction between the two species [125].

## 6. Copper Oxide and Its Derivatives for CO_2_ Gas Sensing

This section summarizes the literature on copper oxides and material combinations thereof for CO_2_ gas sensing. The first part focuses on the sensing of CuO integrated into the bulk, and the second part provides an overview of the use of copper oxide with surface-loaded additives. The calculated sensor responses with relative values refer to Equation (6). The percentages are calculated with Equations (7)–(9). The work function changes are given in mV and calculated by Equation (10). The concentrations of the components of the different compositions are expressed in weight percent (wt%), atomic percent (at%) or mole percent (mol%), and these units refer to the total mass of the mixture. The exposure levels are indicated in accordance with the relevant literature, expressed in either parts per million (ppm) or standard cubic centimeters per minute (sccm). A summary of the sensing performance of all material combinations of CuO is given in Table 3. Not all authors specified the gas concentrations in the same manner in terms of ppm. Therefore, it was not possible to convert all values to the same consistent unit (ppm), and the table also contains L/min and sccm.

### 6.1. CuO

As described in the previous sections, simple and cost-efficient fabrication methods make CuO attractive for gas sensing applications. In 2006, Samarasekara et al. [128] showed that sputtered CuO changes the resistance by about several ohms during exposure to CO_2_.

Resistive sensors

Thermal evaporated copper films that were oxidized are already sensitive to CO_2_ at room temperature. It is reported that the thickness of these pure CuO films needs to be above 200 nm in order to reach a sensor response of 14%; smaller thicknesses such as 100 or 150 nm do not react with CO_2_ at room temperature [74]. It is also possible to use nanostructured CuO by thermal evaporation. CuO nanowires with a diameter of 10 to 50 nm and a length of 10 µm at 400 °C showed a sensor response to CO_2_ of 1.01 [73]. There was also a study on the use of spray pyrolysis as a fabrication method [75]. The film thickness of 2 µm was much larger compared to others. The sensor was exposed to 100 ppm of CO_2_ at room temperature, and the authors reported a sensor response of about 3.5. In contrast, Bhowmick et al. [129] used the sol–gel method to produce a 250 nm thin film of CuO. The required temperature was above 250 °C; below this temperature, the sensors did not react to CO_2_. Due to the dependency on temperature, by increasing the temperature from 250 to 300 °C, the response increased from 10% to 21%. The highest measured sensor response was 114% for a gas pulse of 39,300 ppm of CO_2_ at 300 °C. Relative humidity (r.h.) was not defined in this experiment.

Work function sensors

Drop coating has been used as a deposition method for fabricating the sensing layer of CuO-NP. The influence of humidity, temperature and layer thickness on the work function change was investigated. A study on the influence of layer thickness reported that a larger thickness of about 100 µm led to an improvement in the change in work function compared to a thinner CuO layer of 3 µm. For a 100 µm layer during 4000 ppm CO_2_ exposure, the reported change was 42 mV (dried air) and 97 mV (20% r.h.). The thin layer showed a lower change of 15 mV (dried air) and 41 mV (20% r.h.). In general, the change in the work function was increased by a thicker layer and reached saturation at 60 µm, after which no further increase was observed. The effect of temperature was also considered, with an increase from room temperature to 110 °C. The measurement indicated that the change in the work function (86 mV) at room temperature reached a maximum at 50 °C (130 mV) and then decreased at 65 °C (95 mV) at 45% r.h. with 4000 ppm CO_2_. Higher temperature reduced the change massively, as illustrated in Figure 10 [130]. Humidity also significantly changed the work function behavior. The measurements were performed at room temperature and with 4000 ppm pulses of CO_2_. In dried air, the work function change was the lowest at 42 mV. The largest change in the work function, 91 mV, occurred at 30% r.h.; with an increase to 80% r.h., the change was reduced to 52 mV [79].

### 6.2. BaTiO_3_/CuO

The easy methods of fabricating BaTiO_3_ make it interesting for gas sensing applications. BaTiO_3_ is synthesized by a solid-state reaction of a mixture of BaCO_3_ and TiO_2_. After a deposition technique such as screen printing is used, the layer is sintered [131]. Pure BaTiO_3_ is already sensitive to CO_2_, with a high sensor response (2.9) to 5000 ppm [132]. The reversible reaction of BaTiO_3_ with CO_2_ leads to the formation of BaCO_3_ and TiO_2_. Also, the combination of BaCO_3_ and water causes a reaction with CO_2_ that results in Ba(HCO_3_)_2_ [133]. The combination of BaTiO_3_ and CuO is under research to improve the sensor behavior. The sensing mechanism is based on a p–n heterojunction of BaTiO_3_ (n-type) and CuO (p-type). Bulk ceramic, thick film and thin film are the three types of BaTiO_3_/CuO in use [134,135].

Resistive sensors

Another technique to produce BaTiO_3_/CuO thin films is RF sputtering. In [136], the thickness varied between 150 and 500 nm. The highest response, around 35% at an operating temperature of 250 °C, was achieved with a film 250 nm thick, whereas with thicker films, the bulk conductivity overrules the surface influence [136]. Zheng et al. [131] studied a sintered BaTiO_3_/CuO pellet in which BaTiO_3_ and CuO were mixed at a molar ratio of 1:1. After sintering and being connected to electrodes, the pellet was tested against 5000 ppm of CO_2_. At an optimum temperature of 420 °C, a sensor response of 1.26 was achieved.

Impedance sensors

Mandayo et al. [137] fabricated an RF sputtered BaTiO_3_/CuO film on an alumina substrate with a Pt heater. The authors reported that the linear behavior of the sensor response to the concentration was between 500 and 2000 ppm CO_2_. At 5000 ppm of CO_2_, the sensor reached saturation, with a response above 6% at 300 °C with 40% relative humidity. The sensor was also tested under relative humidity levels between 40 and 80%, and the results demonstrated that the effect caused by humidity was not relevant.

#### 6.2.1. BaTiO_3_/CuO Doped with Metals and Metal Oxides

Resistive sensors

The influence of doping metals such as Au, Ag, Pt, Pd, Ce, Mg, Sr, La, Zn, Fe and Bi was studied by Zheng et al. [131]. BaTiO_3_ and CuO were mixed at a molar ratio of 1:1 and different doping metals were added. After being sintering and connected to electrodes, the pellet was tested against 5000 ppm of CO_2_. It was shown that doping with metal oxides such as Fe_2_O_3_ and CeO_2_ reduced the sensor response, while ZnO, Bi_2_O_3_, SrO and La_2_O_3_ increased the response. However, a much higher operating temperature was needed, over 550 °C. The best sensor response was 1.59, which was achieved with doping with 1 mol% Ag. A variation of the Ag content was tested, resulting in increased sensor response to 1%, with a constant level of 1–3% and a decrease larger than 3%. At a concentration of 6 mol% Ag, the sensor no longer reacted to CO_2_. In contrast, Rudraswamy et al. [136] used RF sputtering to produce BaTiO_3_/CuO thin films with different doping concentrations of silver. The response of the sensor reached the maximum with 1 wt% silver. With higher silver content, the conductivity increased and, therefore, the response decreased. At the best operating temperature of 250 °C, the observed response was 14% for 350 ppm, 59% for 500 ppm and 70% for 1000 ppm.

Impedance sensors

Shwetha et al. [76] used RF sputtering to produce a 150 nm BaTiO_3_/CuO doped with 1% Ag. The sensor was operated at 300 °C and reached a sensor response of 70% (resistance based) and 95% (capacitance based) with 1000 ppm of CO_2_. Notably, a pioneer in combining BaTiO_3_ with CuO is Jaime Herrán [134,135,137,138,139], who studied a multilayer stack consisting of BaTiO_3_/CuO with Ag layers in between. The film stack consisted of four layers of BaTiO_3_/CuO at an equimolar ratio separated by silver layers, with a resulting thickness of 35 nm, on an alumina substrate. The Ag concentration was varied between 0 and 4.52% to again determine the optimum. In contrast to the work of Rudraswamy [134], here, the highest response of 18% (resistive based) and 27% (capacitive based) was obtained with Ag content of 2.26 at% for 5000 ppm CO_2_ at an operating temperature of 300 °C with 40% relative humidity.

A possible reaction mechanism was determined by diffuse reflectance infrared Fourier transform spectroscopy (DRIFT) analysis conducted by Herran et al. At the surface, BaCO_3_ coexists with BaTiO_3_ and can also be formed by the reaction of CO_2_ with oxygen (Equation (18)) [133,140,141]:(18)$${\mathrm{CO}}_{2\left(\mathrm{gas}\right)}+1/2O_{2\left(\mathrm{gas}\right)}+2e_{\left(\mathrm{metal}\right)}^{-}↔{\mathrm{CO}}_{3}^{2-}$$

In the presence of water, the carbonates on the surface reversibly react to bicarbonates (Equation (19)):(19)$${\mathrm{BaCO}}_{3}+H_{2}O+{\mathrm{CO}}_{2}↔\mathrm{Ba}{\left({\mathrm{HCO}}_{3}\right)}_{2}$$

#### 6.2.2. BaTiO_3_/CuO Doped LaCl_3_ or La_2_O

Impedance sensors

Lee et al. [49] produced BaTiO_3_/CuO with LaCl_3_ or La_2_O_3_ by using the powder mixing synthesis method. After screen printing and sintering, the sensors were tested against 10,000 ppm of CO_2_ at different temperatures from 400 to 650 °C. With pure BaTiO_3_/CuO, a sensor response of 1 was achieved. The highest sensor response of around 1.28 was reached when 10 wt% LaCl_3_ was used as additive. The use of 10 wt% La_2_O_3_ slightly increase the sensor response up to 1.05. Another interesting detail is that using 10 wt% of BaTiO_3_/LaCl_3_ without CuO caused a much higher sensor response of 1.58. This example shows that not all combinations using CuO can be improved by additives.

### 6.3. Cu/Fe Oxides

The good mechanical, electrical and magnetic properties of ferrites are well known in several applications of materials for electrical engineering. The group of ferrites is based on Fe_3_O_4_ (magnetite), a mixed-valency compound containing Fe^2+^ and Fe^3+^ ions. These ions can be easily substituted by other transition metal cations, denoted as M, to form M_x_Fe_3-x_O_4_ compounds. Fe_3_O_4_ crystallizes in a spinel structure, in which the oxygen ions resemble a face-centered cubic sublattice (fcc) with octahedrally or tetrahedrally coordinated cation sites. The site preference is mainly given by the d-electron configuration of the cations. Usually, divalent cations occupy the tetrahedral sites and trivalent cations occupy the octahedral sites (as in the mineral “spinel” MgAl_2_O_4_), representing a so-called normal spinel. In Fe_3_O_4_, the Fe^2+^ shows a strong preference for octahedral sites. Hence octahedral sites are occupied by Fe^2+^ and Fe^3+^ cations, and tetrahedral sites are occupied by Fe^3+^ cations. This configuration is called an inverse spinel. Ferrites with M = Zn, Cd or Mn form a normal spinel structure. Others with M = Ni, Co, or Cu and Mg preferentially form an inverse spinel structure [142]. Many ferrites with M = Cd, Co, Cu, Mg, Ni or Zn can be used for gas sensing applications. Many different gases can be measured with these ferrites, including carbon monoxide, methane, ethyl alcohol, oxygen, hydrogen sulfide, chloride and liquefied petroleum gas (LPG), as well as carbon dioxide [143]. The use of copper in ferrites leads to compounds that belong to the n-type semiconductors. Often the combination of Cu_x_Fe_3−x_O_4_ and CuO is used to form p–n heterojunctions, which increases the gas sensing properties [86]. Some papers have also reported the gas sensing performance of copper ferrites against CO, H_2_, LPG, C_2_H_2_ and H_2_S, but only a few papers have reported the sensing performance with CO_2_ [86,143].

Resistive sensors

Sumangala et al. [144] reported on a CuO/CuFe_2_O_4_ sensor with a maximum sensor response to CO_2_ of 10%. The CuO/CuFe_2_O_4_ composite was prepared via co-precipitation with oxalate precursor. They tested different compositions by varying the copper oxide content from 0 to 100% against carbon dioxide. The gas measurement was performed in dried air (0% humidity) using 5000 ppm CO_2_ with temperatures from 200 to 400 °C. The sensor showed the maximum response at 350 °C, which was determined as the optimum operating temperature. The highest response was achieved with pure copper ferrite with zero CuO content. They also reported that the amount of copper oxide did not increase the sensing of CO_2_ gas.

Impedance sensors

The impedance sensing behavior of RF sputtered CuO and Cu_x_Fe_3−x_O_4_ was studied by Chapelle and Oudrhiri-Hassani. A 50 nm thick film was fabricated and measured in dried air [77,86,145]. Oudrhiri-Hassani et al. determined that the optimum temperature was about 200 °C, at which they achieved a response of about 27% using 5000 ppm CO_2_ [77]. In contrast, Chapelle et al. defined the optimum operating temperature as 250 °C, with responses of 40% [145] and 50% [86] using 5000 ppm CO_2_. They also found that increasing the film thickness to 300 nm caused a lower response, around 15%, at an ideal operating temperature of 370 °C [145].

### 6.4. Ni_x_Cu_1−x_Fe_2_O_4_

Few papers report the gas sensing of Ni_x_Cu_1−x_Fe_2_O_4_. However, Ni/Cu ferrites are used in some fields, such as power applications and electromagnetic interference filtration, because of their good magnetic properties. In the field of gas sensors, Ni_x_Cu_1−x_Fe_2_O_4_ is very sensitive to SO_2_ and ethanol and less sensitive to LPG, acetone and H_2_S [146,147].

Resistive sensors

The CO_2_ gas sensing behavior of Ni_x_Cu_1−x_Fe_2_O_4_ thin film was studied by Singh et al. They produced films with different Ni contents via the sol–gel method. The Ni content was varied between x = 0 and x = 0.8. The best sensor response of 1.17 was achieved with a Ni content of x = 0.8 using 2000 ppm CO_2_ at room temperature in dried air. A lower concentration of Ni led to a lower sensor response [148].

### 6.5. Zn_x_Cu_1−x_Fe_2_O_4_

Zinc ferrite (ZnFe_2_O_4_) is under investigation for gas sensing due to the tuneability of its chemical and physical properties. Its morphology can be varied from nanomaterials (e.g., nanoparticles or nanotubes) to thick films, which makes ZnFe_2_O_4_ a promising material for gas sensing applications [149]. In a review article, Wu et al. [150] noted that ZnFe_2_O_4_ composite based gas sensors made the measurement of H_2_S, acetone, n-butyl alcohol and triethyl-amine possible. Combining it with copper leads to Zn_x_Cu_1−x_Fe_2_O_4_ compositions, which can serve as possible gas sensors for SO_2_, LPG, ethanol and acetone [151,152].

Resistive sensors

Singh et al. [153] also tested the CO_2_ gas sensing of Zn_x_Cu_1−x_Fe_2_O_4_. Here, similar to Ni, the Zn content was increased from 0 to 0.8 for thin films that were fabricated by the sol–gel method. The highest sensor response of 2 (calculated according to Equation (6)) or 114% (calculated according to Equation (7)) was achieved by using x = 0.8 with 2000 ppm CO_2_ at room temperature in dried air.

### 6.6. CuO/ZnO

The combination of copper oxide (p-semiconductor) with zinc oxide (n-semiconductor) leads to p–n heterojunctions at the interface of the two materials. The resulting electron depletion layer and the narrowing of the barrier height serve to enhance the charge transfer. The n-semiconductor film undergoes a huge change in resistance if the p–n junction interferes with a gas. Therefore, the combination of these two oxides can increase the sensitivity to different target gases. Ren et al. [154] produced sensors with a sea urchin CuO/ZnO porous nanostructure and a ZnO porous nanostructure. The sensor containing copper showed higher responses to acetone, formaldehyde, methanol, toluene, isopropanol and ethanol. Other researchers showed that it is also possible to measure carbon dioxide with these sensors [155,156,157,158].

Resistive sensors

The combination of a bilayer system of both materials is applicable for gas sensing, with a reported response of 47% at 375 °C [158]. Tanvir et al. [159] produced a gas sensing layer by drop coating CuO-NP in a ZnO_2_ matrix. The chosen molar ratio was 1:10 CuO/ZnO_2_. After an annealing step, the ZnO_2_ was converted to ZnO, forming a CuO/ZnO nanoparticle system. This sensing material was then exposed to 1000, 2000, 3000 and 4000 ppm of CO_2_. The relative humidity was set to 30% and the material was heated to 300 °C. A sensor response of 12, 20, 30 and 35% was achieved for the respective concentrations of CO_2_. Changes in humidity were also investigated. The sensor response increased from dried air to 30% r.h., whereas higher levels of humidity enormously decreased the sensor response. Additionally, alternative structures, such as ZnO/CuO nanorods, have been employed for the purpose of gas sensing. The hydrothermal process leads to the growth of hierarchical ZnO/CuO nanorods on a ZnO seed layer. With these nanorods, a response of 9.7% was achieved for 1000 ppm of CO_2_ at room temperature [156]. Doping of ZnO/CuO nanoflowers with Ag was also investigated. It was reported that the doping concentration plays an important role in the gas sensing performance. When the concentration was increased from 1 wt% to 5 wt%, the sensor response reached the maximum of 18.4% at 2 wt%, with the measurement taken at room temperature with 52% relative humidity and 1000 ppm CO_2_. Doping of nanoflowers with 2 wt% Ag was also tested against different concentrations of CO_2_. There was a linear correlation between the sensor response and the concentration range of 150 to 1000 ppm CO_2_. The sensor response increased from 7.53% to 18.4%. The influence of humidity was studied at 32%, 52%, 62% and 72%, and a small increase in the response from 14.8% to 15.9% was detected [160].

Work function sensors

Musa et al. [155], in their work, fabricated a ZnO/CuO composite thin film sensor for carbon dioxide at room temperature. Films with different precursor solutions of Zn/Cu with a molar ratio of 0.3, 0.5 and 0.7 were fabricated by sol–gel dip-coating on a glass substrate. The sensors were then tested against 100, 500, 1000 and 10,000 ppm of carbon dioxide. Helium was used as the carrier gas, and the relative humidity was constant at around 78%. The highest response (48%) was achieved for ZnO/CuO composite with a molar ratio of 0.7, which means that a higher ZnO fraction leads to a higher response.

### 6.7. CuO/TiO_2_

TiO_2_ is also a subject of research for its potential as a gas sensing material due to its high thermal and chemical stability. Rao and Roy showed that nanotube arrays made from electrochemically anodized TiO_2_ can be stabilized by means of a solvothermal treatment to withstand temperatures up to 800 °C [161]. The use of Cr-doped TiO_2_ allows the measurement of CO_2_ at room temperature [48]. The combination of CuO and TiO_2_ causes p–n heterojunctions. CuO/TiO_2_ composite has been shown to respond to gases like NO_2_, H_2_, CO and ethanol at operating temperatures between 200 and 300 °C [162].

Resistive sensors

It is already possible to measure CO_2_ at room temperature, which was shown by Mude [163]. Different combinations of CuO and TiO_2_ from 0 to 100% were used to identify the best combination. Thick film was prepared via screen printing on a 7.5 × 2.5 cm^2^ glass substrate. The results indicate an increase in sensor response with increasing TiO_2_ content, and the maximum was reached with a combination of 40% TiO_2_ and 60% CuO.

### 6.8. CuO/SnO_2_

As the first metal oxide, SnO_2_ has been studied as a gas sensing material, and nowadays around 27% of the research on gas sensors is performed on this material. SnO_2_ is the most studied metal oxide for gas sensing applications due to its high chemical and thermal stability. Therefore, it is not surprising that there are commercial sensors containing SnO_2_ on the market [164]. Pure SnO_2_ can be used for the detection of VOCs such as ethanol and acetone. There are many studies, which are reviewed in [165], that show the influence and positive effects of different doping materials with SnO_2_ in detecting VOCs. SnO_2_ nanowires used as sensing material can selectively detect NO_x_, H_2_ and CO. The combination of SnO_2_ with CuO shows good sensor behavior for the detection of H_2_S. Here, an n-type semiconductor (SnO_2_) is combined with a p-type semiconductor (CuO), leading to p–n heterojunctions [166].

Resistive sensors

Sol–gel synthesis is the most used method for fabricating SnO_2_ and CuO composite. The precursor of tin is often tin chloride (SnCl_4_), whereas for copper nitrate (Cu(NO_3_)_2_), copper chloride (CuCl_2_) or cupric (II) acetate monohydrate (Cu(CO_2_CH_3_)_2_H_2_O) can be used. After an annealing step, the mixture can be used directly. There is also a special molar ratio needed for both materials to react to CO_2_. Xu et al. [167] found that CuO/SnO_2_ at a molar ratio of 1:8 and a particle size smaller than 20 nm showed a change in resistance to CO_2_. Higher or smaller fractions of SnO_2_ did not show a reaction to CO_2_. Zhang et al. [168] fabricated a gas sensor with CuO/SnO_2_ at a molar ratio of 1:1 and investigated the influence of particle size on the gas sensing performance. Increasing the annealing temperature changed the average particle size from 20 nm (600 °C) to 6 µm (1200 °C). It was reported that a smaller particle size improved the sensor response, caused by a stronger influence of the gas interactions on the surface. The SnO_2_/CuO was tested in a range of 300,000 to 1,000,000 ppm of CO_2_ at 400 °C in an argon atmosphere. The sensor response increased from 1.8 to 3.1. The authors of [169] used CuO/SnO_2_ at a molar ratio of 1:1 mixed with 0.5 wt% Ag. The sensor response to CO_2_ strongly increased with the noble metal compared to pure CuO/SnO_2_. The measured response of the doped sensitive layer to 10,000 ppm of CO_2_ at an operating temperature of 300 °C was 72%.

### 6.9. Cu_x_O/NiO

Nickel is often used as catalyst for chemical reactions. Doping of CuO with NiO for gas sensing was described by Vijayakumari et al. [170]. The radius of copper ion (II+) is 0.57 A and that of nickel (II+) is 0.55 A, which means that there is no lattice distortion by doping. Doping with nickel changes the band gap and enhances the electrical behavior of copper. Furthermore, it functions as a catalyst on the surface, facilitating an increase in carbon dioxide dissociation as a result of doping with metal oxides. The combination of NiO and CuO forms a p–p heterojunction, where a hole depletion layer is formed at the NiO side and a hole accumulation layer at the CuO side [127].

Resistive sensors

As reported in [170], polyethylene glycol (PEG) can be used to modify and control the size of Cu_x_O nanostructures. Therefore, a nanosystem of CuO with PEG was formed and doped with 10 wt% NiO. A response of 30% was reached for 1000 ppm CO_2_ at an operating temperature of 250 °C.

### 6.10. Graphene

Graphene consists of a monolayer of carbon atoms that are densely packed in a honeycomb crystal lattice. Sp2-hybridized carbon atoms are arranged in a two-dimensional planar sheet [171]. This nanostructured material has a large surface area and intrinsic electrical properties that makes it very sensitive to changes in the surrounding atmosphere. Graphene has very high thermal conductivity, good electrical conductivity, high mechanical strength and a high adsorption coefficient. Therefore, graphene attracts more interest than other carbon systems like fullerenes (0D), nanotubes (1D) or graphite (3D). The easier integration of graphene in microfabrication techniques is one advantage. Another advantage of graphene is its good electrical and mechanical properties, which enable gas measurement at room temperature [172,173]. Pure graphene interacts with carbon dioxide via chemisorption and physisorption [174]. The use of a dopant like boron, nitrogen or silicon supports carbon dioxide adsorption on the surface. Graphene doped with Ti shows high sensitivity to CO, NO, SO_2_ and HCHO. The use of copper increases the sensitivity to H_2_S and CO_2_ [171,175].

Resistive sensors

Keerthana et al. [175] fabricated a graphene oxide/cupric oxide sensor. The substrate was made of glass 15 × 15 mm in size and 1.1 mm thick. A mixture of graphene and cupric oxide was spin-coated on the substrate. After annealing, the material was electrically connected by using a silver paste. The sensor was tested against 250, 500 and 750 ppm CO_2_ at an operating temperature of 100 °C, achieving a response of 646, 885 and 963%, respectively. However, the test was performed under dried air; therefore, it did not show the influence of humidity.

### 6.11. CuO/Ba

As described in the previous section, combinations with BaTiO_3_ are widely studied. There is also the approach of combining CuO with barium. Doping CuO with Ba^2+^ changes the band structure compared to the bare CuO material [176].

Resistive sensors

The successive ionic layer adsorption and reaction (SILAR) technique was used for the deposition of CuO with 4 and 6 mol% Ba [176]. The parameters for the gas measurement setup were room temperature and gas flow slowly increasing from 20 to 100 sccm. The sensor response of CuO doped with 4 mol% Ba reached 1.79% (20 sccm) and 2.97% (100 sccm). CuO with 6 mol% Ba was much better, with the sensor response increasing from 4.1% to 9.4% for 20 and 100 sccm, respectively.

### 6.12. CuO/Na

The combination of CuO with Na was demonstrated to be an efficient CO_2_ gas sensing material in [177].

Resistive sensors

It is also possible to fabricate gas sensors based on CuO doped with 2% Na with the SILAR technique. A fabricated sensor was tested at room temperature and the flow rate was increased from 20 to 100 sccm, which resulted in an increase in the sensor response from 5.2% to 12.8%.

### 6.13. CuO/Au

The gas sensing ability of CuO with surface-loaded Au is also under investigation. The authors of [178] reported that CuO/NWs in combination with sputtered Au on the surface was sensitive to gases including CO, NO_2_, C_7_H_8_ and C_6_H_6_.

Resistive sensors

Wimmer-Teubenbacher et al. [179] fabricated a CuO thin film with small structures of Cu using e-beam lithography. After thermal oxidation, the structures grew together and formed a continuous CuO thin film. The final film was tested at 300 °C (Figure 11). However, the authors observed the highest sensor response of 32% at 350 °C and 50% r.h. during exposure to 2000 ppm CO_2_. The sensing layer was then functionalized with Au-NPs by the drop coating method. The operating temperature was reduced to 300 °C, and the sensor response reached 365% for 2000 ppm CO_2_ with 50% r.h.

**Table 3 sensors-24-05469-t003:** Copper-based materials for sensing of CO_2_ gas.

Type	Morphology		Method	Operating Temperature (°C)	Relative Humidity (%)	CO_2_ (ppm)	Sensor Response	Equation	References
CuO									
Resistive sensor	Thick film 12.7 µm		Sputtering	160	Not defined	Not defined	5.1	Not defined	[128]
	Thin film 200 nm		Thermal evaporation	25	Not defined	Not defined	14%	$S=\frac{R_{g}-R_{a}}{R_{a}}$	[74]
	Thick film 2 µm		Spray pyrolysis	25	Not defined	100	3.5%	$S=\frac{I_{g}-I_{a}}{I_{a}}$	[75]
	CuO nanowires 10–50 nm diameter		Thermal evaporation	25	Not defined	5 L/min	1.01	$S=R_{g}/R_{a}$	[73]
	Thin film 240 nm		Sol–gel	250	Not defined	8354	10%	$S=\frac{R_{g}-R_{a}}{R_{a}}$	[129]
				275			15%		
				300			21%		
	Thin film 240 nm		Sol–gel	300	Not defined	39,300	114%	$S=\frac{R_{g}-R_{a}}{R_{a}}$	[129]
Work function sensor	CuO-NP		Drop coating	25	45	4000	86 mV	$\mathrm{CPD}=\frac{1}{e}*\Delta \Phi$	[130]
					60		68 mV		
				50	45		130 mV		
				65	45		95 mV		
	CuO-NP 100 µm		Drop coating	25	0	4000	42 mV	$\mathrm{CPD}=\frac{1}{e}*\Delta \Phi$	[79]
					30		91 mV		
					45		86 mV		
CuO									
Work function sensor	CuO-NP 100 µm		Drop coating	25	0	4000	42 mV	$\mathrm{CPD}=\frac{1}{e}*\Delta \Phi$	[180]
					20		97 mV		
	CuO-NP 3 µm		Drop coating	25	0	4000	15 mV	$\mathrm{CPD}=\frac{1}{e}*\Delta \Phi$	[180]
					20		41 mV		
BaTiO_3_/CuO									
Resistive sensor	Thin film		RF sputtering	250	Not defined	500	35%	$S=\frac{R_{g}-R_{a}}{R_{a}}$	[136]
	Sintered pellet		Sintering	420	Not defined	5000	1.26	$S=R_{g}/R_{a}$	[131]
Impedance sensor	Thin film		RF sputtering	300	40	5000	6%	$S=\frac{R_{g}-R_{a}}{R_{a}}$	[137]
BaTiO_3_/CuO doped with Ag									
Resistive sensor	Sintered pellet 1% Ag		Sintering	430	Not defined	5000	1.59	$S=R_{g}/R_{a}$	[131]
	Thin film 1% Ag		RF sputtering	250	Not defined	350	14%	$S=\frac{R_{g}-R_{a}}{R_{a}}$	[136]
						400	21%		
						450	38%		
						500	59%		
						1000	70%		
BaTiO_3_/CuO doped with Ag									
Impedance sensor	Multilayer 35 nm BaTiO_3_/CuO with silver layers between	0 (at% Ag))	RF magnetron sputtering	300	40	5000	9%	$S=\frac{R_{g}-R_{a}}{R_{a}}$	[134,135]
		1.5 (at% Ag)					14.5%		
		2.26 (at% Ag)					18%		
		4.52 (at% Ag)					11%		
	Multilayer 35 nm BaTiO_3_/CuO with silver layers between	0 (at% Ag))	RF magnetron sputtering	300	40	5000	18%	$S=\frac{C_{g}-C_{a}}{C_{a}}$	[134,135]
		1.5 (at% Ag)					25%		
		2.26 (at% Ag)					27%		
		4.52 (at% Ag)					18%		
	Thin film (150 nm)		RF sputtering	300	Not defined	1000	70%	$S=\frac{R_{g}-R_{a}}{R_{a}}$	[76]
	Thin film (150 nm)		RF sputtering	300	Not defined	1000	95%	$S=\frac{C_{g}-C_{a}}{C_{a}}$	[76]
BaTiO_3_/CuO doped with LaCl_3_ or La_2_O_3_									
Impedance sensor	Thick film 2–5 µm with 10 wt% LaCl_3_		Screen printing	400	Not defined	10,000	1.28	$S=R_{g}/R_{a}$	[49]
	Thick film 2–5 µm with 10 wt% La_2_O_3_		Screen printing	400	Not defined	10,000	1.05	$S=R_{g}/R_{a}$	[49]
CuO/CuFe_2_O_4_									
Resistive sensor	Thick film		Co-precipitation (paste)	350	0	5000	10%	$S=\frac{R_{g}-R_{a}}{R_{a}}$	[144]
Impedance sensor	Thin film (50 nm)		RF sputtering	250	0	5000	40%	$S=\frac{Z_{g}-Z_{a}}{Z_{a}}$	[145]
	Thin film (300 nm)		RF sputtering	370	0	5000	15%	$S=\frac{Z_{g}-Z_{a}}{Z_{a}}$	[145]
Cu_x_Fe_3-x_O_4_									
Impedance sensor	Thin film		RF sputtering	200	0	5000	27%	$S=\frac{Z_{g}-Z_{a}}{Z_{a}}$	[77]
	Thin film		RF sputtering	250	0	5000	48%	$S=\frac{Z_{g}-Z_{a}}{Z_{a}}$	[86]
Ni_x_Cu_1−x_Fe_2_O_4_									
Resistive sensor	Thin film		Sol–gel	RT	Not defined	2000	1.17	$S=R_{a}/R_{g}$	[148]
Zn_x_Cu_1−x_Fe_2_O									
Resistive sensor	Thin film		Sol–gel	RT	Not defined	2000	2	$S=R_{a}/R_{g}$	[153]
SnO_2_/CuO									
Resistive sensor	Thin film 20 nm		Sol–gel	400	Not defined	250,000	1.8	$S=R_{g}/R_{a}$	[168]
						500,000	2.2		
						750,000	2.7		
						1,000,000	3.1		
SnO_2_/CuO with 0.5 wt% Ag									
Resistive sensor	Nanospheres		Hydrothermal process	300	Not defined	10,000	72%	$S=\frac{R_{a}-R_{g}}{R_{a}}$	[169]
						100,000	76%		
						250,000	81%		
						500,000	83%		
						1,000,000	92%		
ZnO/CuO									
Work function sensor	Thin film (MR 0.3)		Sol–gel dip-coating	31	78	100	4%	$S=\frac{R_{a}-R_{g}}{R_{a}}$	[155]
						500	10%		
						1000	22%		
						10,000	31%		
	Thin film (MR 0.5)		Sol–gel dip-coating	31	78	100	6%	$S=\frac{R_{a}-R_{g}}{R_{a}}$	
						500	22%		
						1000	38%		
						10,000	42%		
	Thin film (MR 0.7)		Sol–gel dip-coating	31	78	100	16%	$S=\frac{R_{a}-R_{g}}{R_{a}}$	
						500	30%		
						1000	42%		
						10,000	48%		
Resistive sensor	Bilayer thin film		Spin coating	225	Not defined	2500	21%	$S=\frac{R_{g}-R_{a}}{R_{a}}$	[158]
				250			18%		
				275			21%		
				300			28%		
				325			36%		
				350			47%		
	Nanorods		Hydrothermal process	RT	Not defined	150	1.4%	$S=\frac{R_{g}-R_{a}}{R_{a}}$	[156]
						250	2%		
						500	3.8%		
						750	6.1%		
						1000	9.7%		
	Nanoflowers	1 wt% Ag	RF sputtering	RT	52	1000	11.5%	$S=\frac{R_{g}-R_{a}}{R_{a}}$	[160]
		2 wt% Ag					18.4%		
		3 wt% Ag					8.1%		
		5 wt% Ag					4.5%		
	Nanoflowers With Ag 2 wt%		RF sputtering	RT	52	150	7.53%	$S=\frac{R_{g}-R_{a}}{R_{a}}$	[160]
						250	10.6%		
						500	12.46%		
						750	15.2%		
						1000	18.4%		
	Nanoflowers with 2 wt% Ag		RF sputtering	RT	32	750	14.8%	$S=\frac{R_{g}-R_{a}}{R_{a}}$	[160]
					52		15.2%		
					62		15.5%		
					72		15.9%		
	CuO-NPs with ZnO		Drop coating	300	30	1000	12%	$S=\frac{R_{g}-R_{a}}{R_{a}}$	[159]
						2000	20%		
						3000	30%		
						4000	35%		
Graphene oxide/cupric oxide									
Resistive sensor	-		Spin coating	100	Not defined	250	646%	$S=\frac{R_{g}-R_{a}}{R_{a}}$	[175]
						500	885%		
						750	963%		
Cu_x_O-PEG/NiO									
Resistive sensor	Nanostructured matrix (10 wt% Ni-doped)		Microwave grown	250	Not defined	1000	30	Not defined	[170]
CuO/TiO_2_									
Resistive sensor	Thick film		Screen printing	RT	Not defined	300	0.6	Not defined	[163]
						600	2		
						900	2.5		
CuO/NPs									
Resistive sensor	CuO with 4 mol% Ba		SILAR	RT	Not defined	100 sccm	2.97%	$S=\frac{R_{g}-R_{a}}{R_{a}}$	[176]
	CuO with 6 mol% Ba		SILAR	RT	Not defined	100 sccm	9.4%	$S=\frac{R_{g}-R_{a}}{R_{a}}$	[176]
	CuO with 2% Na		SILAR	RT	Not defined	100 sccm	12.8%	$S=\frac{R_{g}-R_{a}}{R_{a}}$	[177]
	CuO with Au		Drop coating	300	50	2000	365%	$S=\frac{R_{g}-R_{a}}{R_{a}}$	[179]

## 7. Toxicity of CuO as a Nanomaterial

The use of copper nanoparticles has raised concerns regarding their toxicity. Exposure to copper oxide nanoparticles (CuO NPs) has been found to generate reactive oxygen species, leading to oxidative stress, inflammation and toxic effects in the respiratory system, gastrointestinal tract and skin [181]. The toxicity of copper nanoparticles is primarily linked to ion dissolution from the surface or nanoparticle destabilization, which can result in DNA damage and cell death [182]. Studies have indicated that the toxic dose of copper nanoparticles is 10 times lower than that of bulk copper, underscoring the importance of evaluating and comprehending the potential risks associated with exposure [183]. In contrast, another study showed that copper oxide nanoparticles are extremely toxic, resulting in nearly 100% cell death, potentially due to DNA damage [184]. Therefore, it is crucial to thoroughly assess the potential risks and toxicity mechanisms associated with the use of copper nanoparticles in different applications.

## 8. Conclusions and Outlook

In this work, we give an overview of different CO_2_ sensor principles based on optical, acoustical and electrochemical mechanisms. Optical sensors are highly sensitive and selective, but they also have limitations. For instance, they are susceptible to interference from water vapor and atmospheric pressure, and they have a limited ability to detect very low gas concentrations because of the required comparatively long interaction path (typically, ~1 cm, and much more for low gas concentrations). Acoustic sensors are of interest due to their broad range of applications, high sensitivity and fast response. However, these sensors have several disadvantages, including poor signal-to-noise performance, complex circuitry and poor reproducibility.

For implementation in smart systems, device miniaturization and low power consumption are key. In this regard, optical and acoustic sensors are significantly limited due to their size, which leaves the main focus on electrochemical sensors. The interest in commercializing gas sensors as high-performance miniaturized consumer electronic devices, therefore, strongly drives the development of advanced SMOx chemiresistive sensors. Especially for future smart home and smart health applications, such as air quality monitoring and breath analysis, highly miniaturized, reliable and sensitive CO_2_ gas sensors are desirable. As is well known, SMOx also has disadvantages, such as poisoning caused by weak acids and sulfur, and, especially, cross-sensitivity with humidity. However, the drawbacks are outweighed by the advantages of SMOx, including very high sensitivity, rapid response and recovery times, and, due to their ability to be miniaturized, the possibility to realize multi-sensor arrays.

In this paper, we also compared different deposition methods for metal oxide thin film fabrication and their huge impact on gas sensing performance. The fabrication method determines the resulting structure and morphology, which can be nanostructures, sintered pellets, or thick or thin films. The resulting structure affects power consumption and sensor response, due to variations in the surface-to-volume ratio. Several studies have highlighted the potential of CuO nanoparticles and thin films in CO_2_ gas sensing applications. Overall, the research progress on copper oxide nanoparticles and thin films in gas sensing indicates a positive trajectory for the advancement of CuO gas sensing technology. Hence, the future outlook of CuO gas sensing appears promising, with ongoing research and development focused on enhancing the 3-S parameters: sensitivity, selectivity and stability.

In our opinion, CuO-based nanostructures (nanoparticles, nanorods, nanowires and nanoflowers) are the most promising for research toward achieving these goals. Not only the gas sensing properties, such as sensitivity, selectivity and response and recovery time, but also the morphology, stability and ability to be integrated in gas sensing devices need to be investigated in order to realize useable and stable sensor devices. In addition, further functionalization with nanoparticles may prove to be a key to optimizing the sensing performance. One of the most important research directions is to further reveal the sensing mechanisms of these types of sensors. However, we are convinced that CMOS-integrated micro-hotplate devices employing CuO nanostructures are the future for efficient CO_2_ sensing, since they hold promise for further miniaturization and ultra-low power consumption (<1 mW).

## Figures and Tables

**Figure 1 sensors-24-05469-f001:**
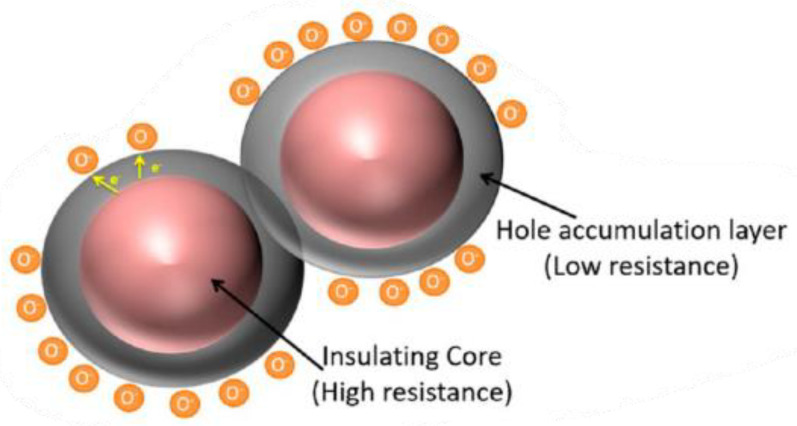
Hole accumulation layer of a p-type semiconductor [66].

**Figure 2 sensors-24-05469-f002:**
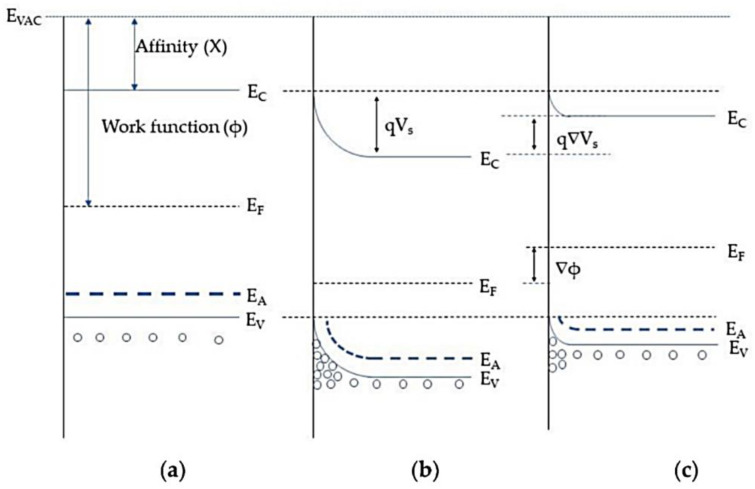
Band model with band bending for p-type semiconductor: (**a**) inert conditions, (**b**) upward band bending and (**c**) decrease of band bending. E_VAC_, energy level of electrons in vacuum level; E_C_, energy level of conduction band; E_F_, energy level of Fermi level; E_A_, energy level of acceptor state; E_V_, energy level of valence band; q, electron charge; qV_S_, potential barrier [69].

**Figure 3 sensors-24-05469-f003:**
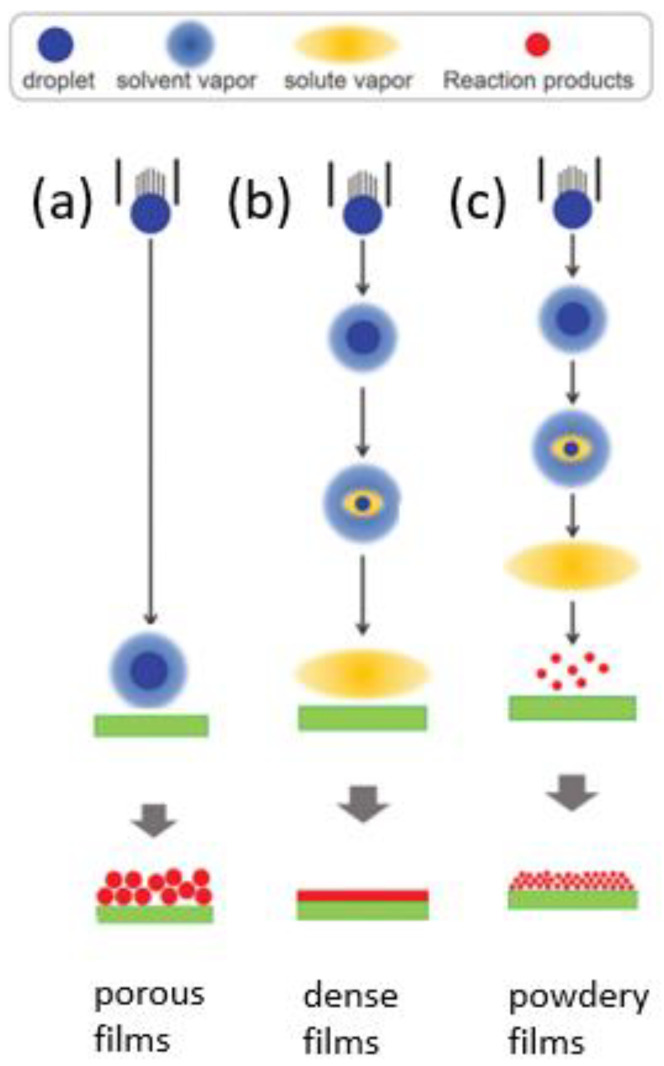
Different types of thin film produced by spray pyrolysis: (**a**) porous films, (**b**) dense films and (**c**) powdery films [95].

**Figure 4 sensors-24-05469-f004:**
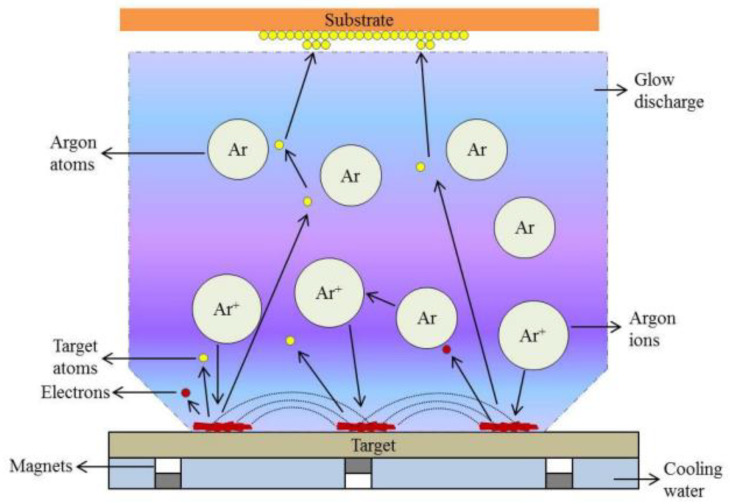
Schematic representation of magnetron sputtering process [99].

**Figure 5 sensors-24-05469-f005:**
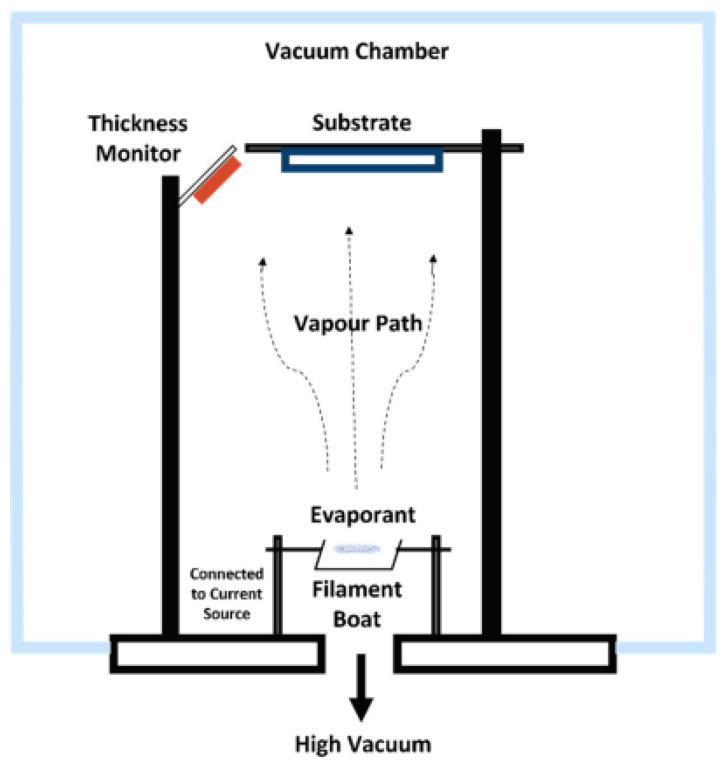
Schematic representation of thermal evaporation process [102].

**Figure 6 sensors-24-05469-f006:**
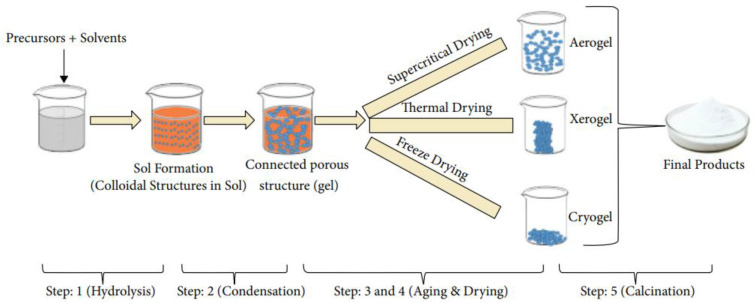
Steps of sol–gel method for fabrication of metal oxides [115].

**Figure 7 sensors-24-05469-f007:**
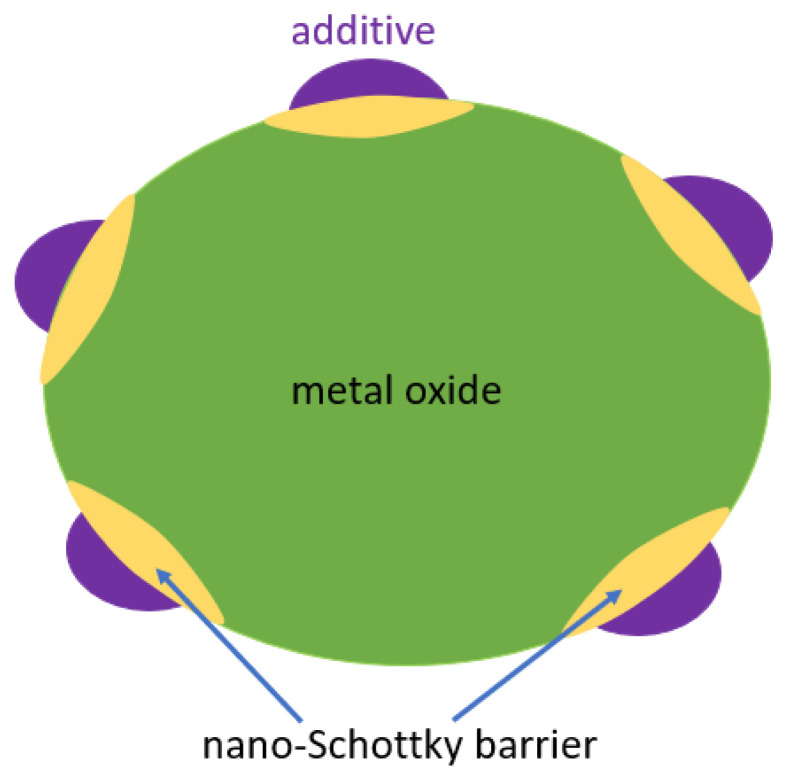
Surface loading of metal oxide with additives.

**Figure 8 sensors-24-05469-f008:**
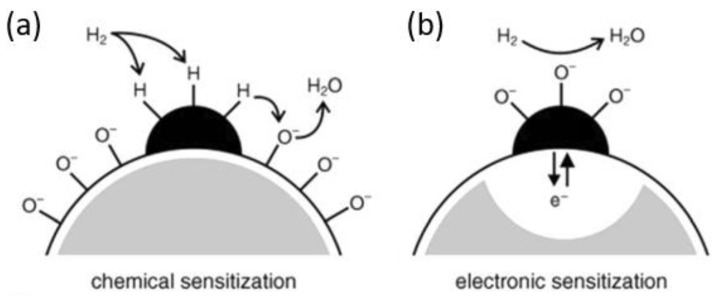
(**a**) Chemical and (**b**) electronic sensitization [123].

**Figure 9 sensors-24-05469-f009:**
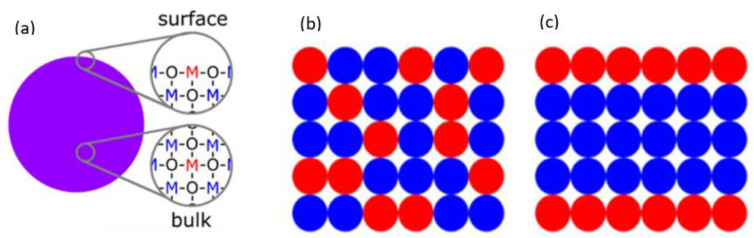
Different metal atoms (blue and red) with (**a**) bulk integration, (**b**) randomly arranged bulk integration and (**c**) multilayer arranged bulk integration [120].

**Figure 10 sensors-24-05469-f010:**
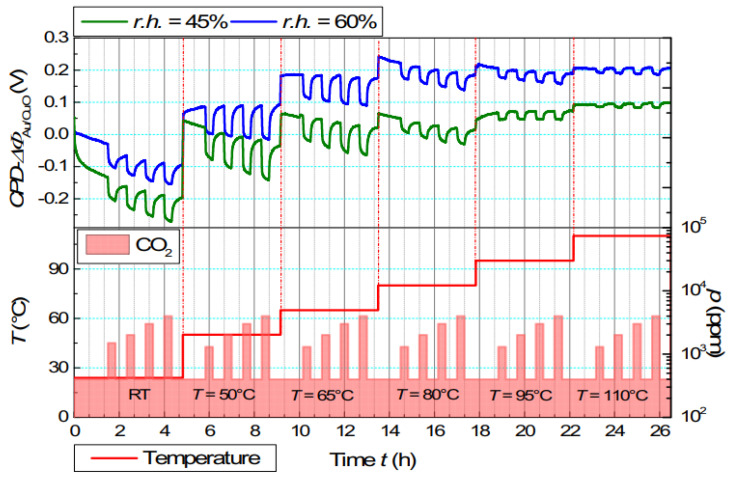
Work function change of CuO-NPs when CO_2_ levels are increased from 400 to 4000 ppm between room temperature and 110 °C, with humidity of 45% (green) and 60% (blue) [130].

**Figure 11 sensors-24-05469-f011:**
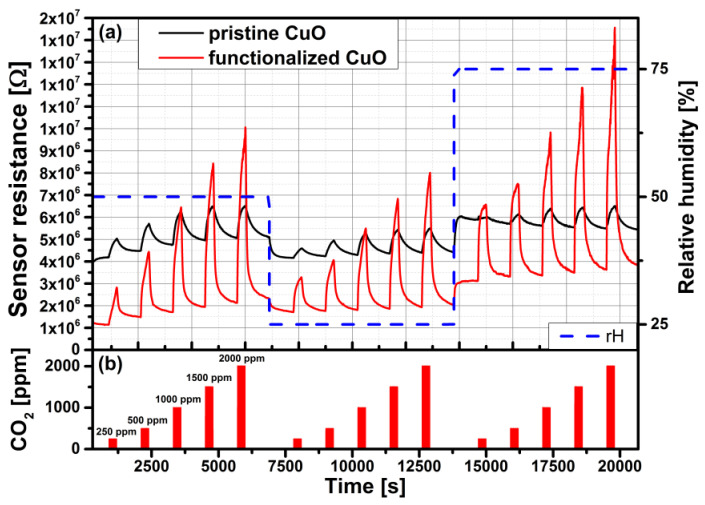
(**a**) Resistance measurement of pristine and functionalized CuO at 300 °C and 25, 50 and 100% r.h.; (**b**) CO_2_ pulses of 250, 500, 1000, 1500 and 2000 ppm [179].

**Table 1 sensors-24-05469-t001:** Summary of health effects caused by CO_2_ at different exposure levels according to the United States Occupational Safety and Health Administration [12] and the Canadian Centre for Occupational Health and Safety [13].

CO_2_ Concentration (ppm)	Health Effects
<1000	No effects (normal indoor value)
1000–2500	Loss of focus and concentration
2500–5000	Tiredness, headache (8 h time weighted average (TWA) of 5000 ppm)
5000–40,000	Increased respiration, headache, fatigue, other symptoms
40,000–100,000	Immediately dangerous to life or health (IDLH), increased heart rate, loss of consciousness
>100,000	Loss of consciousness, risk of death

**Table 2 sensors-24-05469-t002:** Typical precursors for spray pyrolysis of CuO.

Name	Formula	Reference
Copper chloride	CuCl_2_	[90]
Copper (II)—acetate	Cu(CH_3_COO)_2_	[91]
Copper nitrate	Cu(NO_3_)_2_	[92]

## Data Availability

Data are contained within the article.

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
