# Peer review of "A Review of Gas Sensors for CO2 Based on Copper Oxides and Their Derivatives"

_sensors, 2024, doi:10.3390/s24175469_

Round 1

Reviewer 1 Report

Comments and Suggestions for Authors

The review work is focused on a comprehensive description of methods for developing low-cost semiconducting metal oxide (SMOx) gas sensors. The review includes a lot of background data and will be useful for both specialists and beginners in the field of sensor development, as well as specialists from related fields dealing with the problems of air pollution. The wide coverage of the cited articles leaves a good impression of the development of the topic related to SMOx developments, the operating  mechanisms of the main gas sensors and their performances. However, the small number of cited works by the authors themselves is surprising. Examination of the Scopus database shows that the authors have been working on this topic for a long time, I would recommend adding 3-5 of your most significant papers on gas sensors (SMOx) to the citation list. The manuscript is well written and structured (although sometimes there are difficulties in understanding the meaning of some statements), but often the “-” sign in chemical formulas is perceived by the text editor as a hyphen (for example, lines 650, 706, 785, 794). I consider it unacceptable to use a hyphen in chemical formulas (line 578, Ba-TiO3, or line 689 raises doubts about whether Ni-Cu or NiCu was intended). It's better to use a non-breaking hyphen. It is advisable to decipher some abbreviations like MEMS- (l.58), NASICON (l.147), DRIFT (l.631), fss-... (l.651), PEG (l.811) or units of measurement sccm (l.844) at least once. Obviously the problems with reference sources should be corrected. In the abstract of the manuscript, the authors note the importance of intelligent technologies in air quality monitoring, but this is not discussed anywhere else in the review. If the authors wish to reflect the importance of intelligent technologies in their review, I recommend that the authors pay attention to the article (https://doi.org/10.3390/s23136153) and others like it, devoted to the use of artificial intelligence in monitoring indoor air quality. Otherwise, the manuscript abstract should be corrected so that the reader is not misled as to the purpose of the review. The above notes  are not significant and do not require a major revision other than the manuscript abstract.  The manuscript can be published after significant revision of the abstract and/or introduction, as well as some parts of the text.

Some additional notes:

l.47. " To integrate such efficient sensors in smart systems the application of nanotechnology for gas sensing offers several advantages." - Examples (cites) should be given where attempts at such integration have been made.

l.66-68. It is necessary to write more specifically, the phrase "the gas measurements" is too vague.

l.92. In the text "path way" should probably be corrected to "pathway".

l.141-142. " On the other hand there are still challenges like the sensitivity to change of the humidity and high operating temperature (up to 300 °C)."  - Is such a temperature possible in rooms where air monitoring is carried out?

l.194-198. Eqs.2-3. It remains unclear why such a wide temperature range needs to be considered if the introduction is focused on monitoring gas concentrations in buildings and household air pollution. It is necessary to clarify whether the oxide substrate needs to be heated for the sensor to operate? A logical question arises: in which smart gadgets can a gas sensor with an operating temperature of more than 100C be used?

l.205-208. "This means that the conduction of a p-type metal oxide..." - Ri are conductivities or resistances as follows from the figure 2. It is necessary to coordinate the text and figure.

l.237. All designations in Figure 3, such as Ea, Ev, Ef, Ec and Vs must have a description in the figure caption or be mentioned in the text.

l.350-370. Too much scattered information. A detailed drawing will allow the reader to understand the description of the "Sputtering" method much better.

l.412-413. "Studies have been conducted on surfactant-free methods for synthesizing CuO. [92]" - Is there a short conclusion that can be drawn from these studies? I think the reader might find this interesting.

l.419-420. "In the first step called hydrolysis, the solvent, an alcohol or water reacts with the metal alkoxide to end up in a metal hydroxide as shown in Equation 14." - Eq. 14 describes the first stage, it should be explained where the metal alkoxide first appears.

l.454-456. "This combination of a metal oxide with an additive can be explained as nano-Schottky barriers..." - How and what combination can be explained? Unfortunately, Figure 7 does not clarify the issue.

l.472. Why is SnO2 listed in Figure 8 when copper oxides were discussed earlier? If the figure is borrowed from [108], you must indicate permission to use it, otherwise you must draw the original figure.

l.481-482. " The connection of a metal oxide with a noble metal causes a band bending of the conduction band, that results in the formation of a Schottky barrier." - The connection between “a band bending” and the formation of “a Schottky barrier” should be explained by analogy with Figure 3.

l.497. You must specify "a depletion layer" consists of holes or electrons.

l.513-514. "If the grain size is of the additive is smaller than it refers to the surface loading. [103]" - The meaning of this sentence is unclear.

l.543. What is the relationship between sensor percentage response and “a sensor response of 1.01”? A uniform description of sensor characteristics must be used. The rest of the manuscript (l.546, l.593, l.641-644) should also take this into account.

l.615. "The response of the sensor increases up to a maximum at 1 wt%." - What does "1 wt%" mean? The weight percentage has not been seen anywhere before. Some clarification required. Same for "at%" (l.629) and "mol%" (l.845).

l.649-652. Suddenly, the authors began to discuss crystal lattices. "fcc-sub-lattice" - Transcription required. Not everyone reader remembers what types of sublattices exist.

l.703. What does "tick films" mean? Some clarification required.

l.712. What does "The highest sensor response 2" mean? Where is the description of gas sensor 1?

l.718-719. "Also, a depletion layer is formed that leads to a small pathway for the charge carriers." - The sentence is too vague.

l.736. "Some other researchers use different structures for gas sensing." - It's too obvious an expression that doesn't make any sense.

l.741-744. It is necessary to clarify in relation to what the weight percentage is applied.

l.759. What does the phrase “due to its high temperature” mean when applied to a specific substance?

l.769. It is possible to use the phrase "a maximum with a combination of 40% TiO2 and 60% CuO." it would be clearer for the reader.

l.806. Previously, in the description of the mechanisms of operation of gas sensors, nothing was mentioned about the dissociation of carbon dioxide. Why is this noted here, what important role does its dissociation play in the detection of CO2?

l.849. " the gas measuring of CO2" - It is an incorrect phrase.

l.842-855. The use of the "sccm" unit for flow rate makes it difficult to compare the performance of the sensors relative to others described in the review, which are described in terms of concentrations with "ppm" units. Is it possible to bring everything to a single description?

Reviewer 2 Report

Comments and Suggestions for Authors

The manuscript reviews the research progress on CO2 gas sensors based on copper oxides and their derivatives. It provides a detailed introduction to the market applications, sensing principles, and performance characterization of different types of gas sensors. The manuscript discusses the preparation methods of CuO-based gas sensor materials and their performance in CO2 detection. Through the analysis of existing studies, the authors suggest that future research should focus on understanding the reaction mechanisms and doping effects. This manuscript has certain academic value in the field of gas sensor research and development. I would recommend its publication if the following revisions are made:

1. In the first half of the manuscript, such as the Introduction and Market overview and present research sections, it would be beneficial to add more information about copper oxide-based sensors to highlight the research focus of this paper and improve the logical flow.

2. While the manuscript presents many performance parameters of CO2 gas sensors based on copper oxides and their derivatives, it lacks a comparison with other types of sensors. Including such comparisons would provide a clearer view of the advantages and disadvantages of different sensor types.

3. At the end of the manuscript, it is suggested to add a systematic summary of key research points and discuss future research directions in detail, reflecting the authors' insights.

4. Should 114% in line 552 be 114 based on Table 2? The response of 114% in line 713 also appears to contradict the data in Table 2.

5. Lines 80, 339, 469, 523-531, 712 appear with the same statement “Error! Reference source not found.” This is confusing, please make amendments or explanations.

6. In line 24, "the united nations" should be capitalized as "the United Nations". Also, ensure consistent indentation, such as at the beginning of line 444.

Comments on the Quality of English Language

 Minor editing of English language required

Reviewer 3 Report

Comments and Suggestions for Authors

The authors present a well documented review regarding the SMOX-based sensors for CO2 sensing.

The authors should address these issues prior to acceptance:

1. In the Introduction section please insert a table with CO2 (target gas) exposure limits, according to OSHA or NIOSH, and the associated symptoms which occur when the imposed limits are surpassed.

2. Please improve the quality of Figure 3, which is a bit blurry, 

3. Although a Future Outlook section is present in the manuscript, emphasizing the main aspects presented in the paper and the future work it needs to be done, a Conclusion section would be desirable, with more elaborated information about advantages/disadvantages of each sensor category (with examples) discussed in the manuscript. Also some sensor response/recovery plots for selected sensor samples would be nice for the reader to see.

Comments on the Quality of English Language

Please polish the English throughout the entire manuscript, I found errors such as: "In 2022, the united nations declared clean.."-line 24

Round 2

Reviewer 1 Report

Comments and Suggestions for Authors

I would like to thank the authors for their positive feedback and desire to improve the manuscript. I believe that those who read it will now be able to better understand the prospects for the development of SMOx and the possibilities of their integration into smart devices. The corrected manuscript can be published without any additional changes.

Reviewer 3 Report

Comments and Suggestions for Authors

The manuscript was greatly improved after revision.